

# Antibacterial activity and mechanism of sanguinarine against *Providencia rettgeri* in vitro

Qian Zhang[1,2,3,4], Yansi Lyu[1], Jingkai Huang[1], Xiaodong Zhang[1], Na Yu[1], Ziping Wen[1] and Si Chen[3,5]

[1] Department of Dermatology, Shenzhen University General Hospital, Shenzhen University, Shenzhen, China
[2] College of Physics and Optoelectronic Engineering, Shenzhen University, Shenzhen, China
[3] Shenzhen University Health Science Center, Shenzhen, China
[4] Department of Dermatology, PLAGH Hainan Hospital Of PLA General Hospital, Sanya, China
[5] Department of Immunology, Shenzhen University School of Medicine, Shenzhen, China

Corresponding author
Si Chen, chensi@szu.edu.cn

## ABSTRACT

**Background:** Sanguinarine (SAG), a benzophenanthridine alkaloid, occurs in *Papaveraceas*, *Berberidaceae* and *Ranunculaceae* families. Studies have found that SAG has antioxidant, anti-inflammatory, and antiproliferative activities in several malignancies and that it exhibits robust antibacterial activities. However, information reported on the action of SAG against *Providencia rettgeri* is limited in the literature. Therefore, the present study aimed to evaluate the antimicrobial and antibiofilm activities of SAG against *P. rettgeri* in vitro.
**Methods:** The agar dilution method was used to determine the minimum inhibitory concentration (MIC) of SAG against *P. rettgeri*. The intracellular ATP concentration, intracellular pH ($pH_{in}$), and cell membrane integrity and potential were measured. Confocal laser scanning microscopy (CLSM), field emission scanning electron microscopy (FESEM), and crystal violet staining were used to measure the antibiofilm formation of SAG.
**Results:** The MIC of SAG against *P. rettgeri* was 7.8 μg/mL. SAG inhibited the growth of *P. rettgeri* and destroyed the integrity of *P. rettgeri* cell membrane, as reflected mainly through the decreases in the intracellular ATP concentration, $pH_{in}$ and cell membrane potential and significant changes in cellular morphology.
The findings of CLSM, FESEM and crystal violet staining indicated that SAG exhibited strong inhibitory effects on the biofilm formation of *P. rettgeri* and led to the inactivity of biofilm-related *P. rettgeri* cells.

## INTRODUCTION

*Providencia rettgeri (P. rettgeri)* is a common gram-negative bacilli from the Enterobacteriaceae family and is often isolated from wounds, the urinary tract, reptile faces, and human blood (*Wie, 2015*). It may cause gastroenteritis and bacteremia (*Wie, 2015*), which are considered the primary causes of travelers' diarrhea (*Yoh et al., 2005*). *P. rettgeri* is considered a hospital pathogen that often causes urinary tract infections

(*Armbruster et al., 2014*). The first infection of multidrug-resistant *P. rettgeri* in a neonate reportedly caused late-onset neonatal sepsis (*Sharma, Sharma & Soni, 2017*), which was resistant to ampicillin, polymyxins, and first-generation cephalosporins (*Magiorakos et al., 2012*); moreover, the treatment of *P. rettgeri* infection was challenging. In recent, *P. rettgeri* has been the focus of researchers due to the emergence of new multidrug-resistant clinical strain (*Mbelle et al., 2020*), which have threatened public health worldwide. More importantly, at the beginning of the 21st century, *P. rettgeri* caused two food poisoning outbreaks, as reported by the Chinese Center for Disease Control (*Murata et al., 2001*). This report revealed that the risk of exposure to *P. rettgeri* extended beyond individuals who traveled, were were hospitalized, or interacted with the general public.

Biofilms increase food safety risks by increasing the resistance of embedded bacteria to stress that is often encountered in food processing and by acting as a persistent source of microbial contamination (*Giaouris et al., 2014*). Thus, another major reason why *P. rettgeri* causes food contamination is the robustness of bacterial biofilms. Therefore, there is an urgent need to increase the supply of the ideal preservative against *P. rettgeri* in food, preferably a plant extract, given that food safety is crucial.

Sanguinarine (13-methyl(1,3)benzodioxolo(5,6-c)-1,3-dioxolo(4,5)phenanthridinium) (SAG) is an isoquinoline derivative and a benzophenanthridine alkaloid, with a molecular formula of $C_2H_{15}NO_5$ (*Yatoo et al., 2018*). SAG occurs in *Papaveraceas, Berberidaceae* and *Ranunculaceae* families (*Och et al., 2019*). Previous studies have revealed that SAG has antimicrobial, antioxidant, and anti-inflammatory abilities (*Firatli et al., 1994*), as well as cytotoxic effects on soma cancer cells through its action on the $Na^+/K^+$-ATPase transmembrane protein. It is known that $Na^+/K^+$-ATPase with normal physiological functions maintains the resting potential and regulates cellular volume by pumping sodium out of cells and potassium into cells, both against their concentration gradients (*Singh & Sharma, 2018*). SAG is mainly utilized in veterinary clinic to promote the growth of livestock and poultry and prevent diarrhea. Of course, more efforts should be made to further evaluate its effectiveness and safety to meet its clinical application. A previous study reported that SAG possesses antimicrobial acidity against *Psoroptes cuniculi* (*Miao et al., 2012*).

Although the activity of SAG against some bacteria has been reported, to the best of our knowledge, its effect on *P. rettgeri* has not been reported. Therefore, the present study aimed to investigate the antimicrobial and antibiofilm activities of SAG against *P. rettgeri*.

## MATERIALS AND METHODS

### Reagents

Sanguinarine (CAS:5578-73-4; Nanjing, China) was purchased from Nanjing Zelang Meditech Ltd., with a purity level of 98%. For all experiments, 10 mg/mL SAG in dimethyl sulfoxide (DMSO; Sigma–Aldrich, MI, USA) was employed as a stock concentration, and a 10-fold dilution of the stock was prepared in DMSO before addition to culture suspensions to obtain the required SAG concentrations.

## Bacterial and culture condition

*Providencia rettgeri* can form normal biofilms. The clinical *P. rettgeri* strains used in this study included *P. rettgeri*-1, *P. rettgeri*-2, *P. rettgeri*-3, *P. rettgeri*-4, *P. rettgeri*-5, *P. rettgeri*-6 and *P. rettgeri*-7—the seven *P. rettgeri* isolates originally derived from the hydrothorax and urine of patients. The strains were routinely grown in tryptic soy broth (TSB) at 37 °C in a shaking incubator. The cell suspensions were diluted according to 0.5 McFarland standards to prepare inoculum densities of $2 \times 10^6$ colony-forming units (CFU)/mL for the antibacterial activity assay and those of $5 \times 10^8$ CFU/mL for the biofilm assay.

## Minimum inhibitory concentration test

Evaluation of the minimum inhibitory concentration (MIC) was performed by following the broth microdilution (*European Committee for Antimicrobial Susceptibility Testing (EUCAST) of the European Society of Clinical Microbiology & Infectious Dieases (ESCMID), 2000*). The *P. rettgeri* culture was diluted into 96-well plate (Costar, Corning, NY, USA) using TSB at 106 CFU/mL. Tryptic soy agar (TSA) was mixed with SAG with final concentrations ranging from 0 to 500 μg/mL in the culture dish. After mixing, the tested bacteria were spotted on the media and incubated at 37 °C for further 24 h. The MIC value was regarded as the lowest SAG concentration at which no visibility of bacteria growth was sighted. The assays were performed independently thrice.

## Growth curves

The method of *Xu et al. (2017)* for establishing growth curves was slightly modified to assess the antimicrobial effect of SAG on *P. rettgeri*. Then, the log-phase bacterial suspension (100 μL; $OD_{600} = 0.5$) was diluted 100-fold, and treated with SAG at different concentrations of 0, 1/8, 1/4, 1/2, 1 and 2 MIC. Subsequently, the samples were incubated at 37 °C with gentle shaking at 150 rpm. The cultures were sampled at 1 h-intervals and plated on TSA after adequate serial dilutions. Finally, the resulting plates were incubated at 37 °C for 24 h, and then the colonies were manually counted. The dynamic growth or inactivation curves were sets up for the viable cell counts.

## Measurement of ATP levels

The method described by *Sianglum et al. (2018)* was employed to determine the intracellular ATP concentration of *P. rettgeri* with minor modifications. Briefly, *P. rettgeri* suspension ($1 \times 10^5$ CFU/mL) was treated with 0, 1 and 2 MIC of SAG at 37 °C for 1, 2 and 3 h, respectively. Then, the cells were harvested by centrifugation ($8,000 \times g$, 10 min). Simultaneously, the supernatants were filtered and transferred to a test tube to measure the extracellular ATP concentration. The bacterial precipitates were washed twice with 1 mM phosphate-buffered saline (PBS; pH 7.0), suspended in the same PBS, and then placed on ice. Finally, the intracellular ATP was extracted by DMSO. ATP was measured using the ATP Assay Kit (Beyotime Bioengineering Institute, Shanghai, China), according to the manufacturer's instructions.

## Intracellular pH test

The intracellular pH ($pH_{in}$) was evaluated according to the method presented by *Qian et al. (2020)*. 300 μL overnight cultures of *P. rettgeri* were transferred to 30 mL TSB and then incubated at 37 °C for 8 h. After incubation, the cells were centrifuged (8,000×*g*, 10 min), rinsed twice with 50 mM HEPES and five mM EDTA (mixed solution at pH 8.0), and resuspended in 20 mL mixed solution. Subsequently, 3 μM carboxyfluorescein diacetate succinimidyl ester (CFDA-SE; Beyotime Bioengineering Institute) was added to the sample. The sample was incubated at 37 °C for 20 min, then washed once with the mixture solution (pH 7.0; 50 mM PBS and 10 mM MgCl2), and resuspended in the buffer. To eliminate nonconjugated CFDA-SE, glucose (10 mM final concentration) was added and incubated at 37 °C for another 30 min. Cells were then washed twice, resuspended with 50 mM potassium phosphate buffer (pH 7.0), and stored in ice.

Fluorescently stained cell suspensions were treated with SAG (0, 1 and 2 MIC) and transferred into a black opaque 96-well flat-bottomed plate (Nunc, Copenhagen, Denmark, Europe). After treatment for 20 min, fluorescence intensity was assessed at excitation wavelengths of 440 and 490 nm and emission wavelength of 520 nm, where the excitation slit width was 9 nm and the emission were 20 nm. $pH_{in}$ was defined as the ratio of the fluorescence signal at the pH-sensitive wavelength of 490 nm to the fluorescence signal at the pH-insensitive wavelength of 440 nm. The measurements were performed by a fluorescence microplate reader (Thermo Fisher Scientific, Finland, Europe). During the experiment, the system was maintained at 25 °C, and the fluorescence of the control group was measured and subtracted from the value of the treated suspension.

The calibration curves were established for CFDA-SE-loaded cells with mixed solutions of different pH values. The mixture including glycine (50 mM), citric acid (50 mM), $Na_2HPO_42H_2O$ (50 mM) and KCl (50 mM). The pH was adjusted to various values (2–10) with NaOH or HCl, and valinomycin (10 μM) and nigrosine (10 μM) were added to adjust the $pH_{in}$ and $pH_{out}$. Finally, fluorescence intensity was evaluated at 25 °C.

## Determination of membrane potential

The membrane potential was determined according to the method by *Wang et al. (2018)*, with minor modifications. The fluorescent probe bis-(1,3-dibutylbarbituric acid) trimethine oxonol ($DiBAC_4(3)$) was used to determine the changes in bacterial membrane potential. Bacterial suspensions ($1 \times 10^5$ CFU/mL) were centrifuged at 4,000×*g* for 10 min and washed twice with 10 mM PBS (pH 7.0). The cells were treated with different concentrations of SAG (0, 1 and 2 MIC) and incubated at 37 °C for 2 h. Then, the resulting samples were incubated with $DiBAC_4(3)$ at 25 °C for 10 min in the dark and washed once with PBS. The fluorescence microplate reader was utilized to determine the florescence intensity at excitation/emission wavelengths of 492/515 nm. The excitation and emission slit widths were 3 and 5 nm, respectively.

## Evaluation of bacterial membrane integrity

The modified method of *Liu et al. (2018)* was applied to assess the cell membrane integrity of *P. rettgeri* cells using confocal laser scanning microscopy (CLSM; Zeiss LSM 880 with

Airyscan, Bonn, Germany). Overnight cultures were diluted in TSB medium a concentration of $1 \times 10^5$ CFU/mL and treated with SAG (0, 1 and 2 MIC) for 2 h. Subsequently, the cells were harvested by centrifugation at 8,000×$g$ for 5 min, and then resuspended with 10 mM PBS of equivalent volume. Then, the samples were mixed with SYTO 9 and propidium iodide (PI) at 25 °C for 15 min to visualize the cells. Cells were collected and washed with 10 mM PBS to remove excess dyes, and finally measured with a CLSM.

## Transmission electron microscopy

Transmission electron microscopy (TEM) (G2 F20 S-Twin; FEI, Hillsboro, OR, USA) was used to analyze cell morphology as described by *Joung et al. (2016)* with some modification. Overnight cultures of *P. rettgeri* ($1 \times 10^5$ CFU/mL) were diluted and exposed to three concentrations of SAG (0, 1, and 2 MIC) for 4 h at 37 °C. Thereafter, the cultures were centrifuged (10,000×$g$, 8 min), and washed twice with 0.85% NaCl. The suspension was removed, and cell pellets were fixed with 2.5% glutaraldehyde for 12 h at 37 °C. Then, cells were dehydrated through graded alcohols (20%, 50%, 70%, 80%, 90% and 100%) for 15 min; the resulting cell pellets were embedded in resin. Ultrathin samples were incised by ultramicrotome, and uranyl acetate stain was used for TEM.

## Field emission scanning electron microscopy

After treatment with SAG at different concentrations, field emission scanning electron microscopy (FESEM) (Nova Nano SEM-450; FEI, Eindhoven, Netherlands, Europe) findings have revealed changes in the morphology of bacteria, as described previously (*Shi et al., 2018*). The log-phase cells were exposed to SAG at three concentrations (0, 1 and 2 MIC) and incubated for 4 h at 37 °C. The cultures were washed with 10 mM PBS (pH 7.0), and fixed with 2.5% glutaraldehyde for 12 h at 4 °C. The cells were dehydrated by sequential treatment with ethanol concentrations ranging from 30% to 100% for 10 min. Finally, the cells were fixed to the FESEM scaffold, gold sputtering was used to coat the cells under vacuum conditions, and FESEM was used to examine cell morphology.

## Inhibition of the biofilm formation

Effects of SAG on the biofilm formation of *P. rettgeri* were quantitatively analyzed by crystal violet staining method, as described by *Qian et al. (2020)*. *P. rettgeri* cells were incubated in the presence of SAG at 0, 1/32, 1/16, 1/8, 1/4, 1/2, 1 and 2 MIC for 24 h in a 96-well plate. After incubation, the cells washed twice with distilled $H_2O$ to remove the planktonic cells and loosen the adherent cells in the plate. For crystal violet staining assay, the biofilms were stained with 1% crystal violet (100 μL), incubated for 15 min at 28 °C, and washed twice with distilled $H_2O$. Then, the absorbance was measured at 570 nm by a microplate reader. The specific biofilm formation index was determined by attaching and stained bacteria ($OD_{570}$) normalized with the cell growth ($OD_{630}$).

Moreover, the effects of SAG on the biofilm formation were qualitatively evaluated by FESEM and CLSM, as described previously (*Qian et al., 2020*). The slides of the cells were treated with different concentrations at 0, 1/16, 1/8 and 1/4 MIC, incubated for 24 h at

37 °C, washed thrice with 0.85% NaCl, fixed with 2.5% glutaraldehyde in 200 mM phosphate buffer for 4 h at 25 °C, dehydrated with different concentrations of ethanol gradient (30%, 60%, 70%, 80%, 90% and 100%) for 20 min for each concentration, and treated with gold spray. The biofilm formation of *P. rettgeri* was finally subjected to the examination under an FESEM.

*Providencia rettgeri* cells grown on the slides for 12-h were exposed to SAG (0, 1/16, 1/8 and 1/4 MIC) and cultured for 24 h at 37 °C, and then washed twice with 0.85% NaCl. The samples of *P. rettgeri* biofilm were stained with CFDA-SE and then incubated for 15 min at 25 °C for direct visual observation of the biofilm formation using a CLSM. Fluorescence intensity of the cells was measured at excitation/emission wavelengths of 488/542 nm for CFDA-SE.

## Biofilm matrix by CLSM

The changes in major biofilm matrix levels within 24-h-old biofilms of mono or dual species in the presence of SAG were detected by CLSM of the biofilm matrix combined with different dyes. Briefly, biofilms were prepared using the procedure described in the crystal violet biofilm assay section. The resulting samples were washed thrice with 1 mM sterile PBS (pH 7.4) and labeled with three fluorescent dyes: (1) 25% (v/v) SYPRO Ruby, which stains most classes of proteins, and 2 μM SYTO9 dye; (2) 5 μg/mL WGA dye, which labels polysaccharides, and 5 μg/mL water-soluble FM® dyes; and (3) 2 μM PI dye and 2 μM SYTO9 dye. Subsequently, the biofilm was washed twice with 1 mM PBS to remove all planktonic bacteria, where the excitation/emission wavelengths were 450/610 nm for SYPRO Ruby (red), 498/517 nm for SYTO9 (green), 488/617 nm for PI (red), 495/519 nm for WGA (green) and 479/565 nm for water-soluble FM® dyes (red). Color confocal images were observed with CLSM.

## Diffusion bioassay for gatifloxacin within biofilms

To evaluate the diffusion of antibiotics into biofilms, the diffusion of gatifloxacin into biofilms was identified based on the intrinsic fluorescence of CLSM. The biofilms described above were placed on glass coverslips inside the 24-well plate for 48 h at 37 °C in the presence of SAG at 0, 1/16, 1/8 and 1/4 MIC, withdrawn, and gently washed thrice with 10 mM PBS, added with gatifloxacin at a final concentration of 0.4 mg/mL, and further incubated at 37 °C for 4 h. Then, 3 μM SYTO 9 was added and incubated for 15 min to observe gatifloxacin diffusion within biofilms. The samples were washed with 10 mM PBS thrice and observed with CLSM. The emission peak of gatifloxacin at 495 nm was recorded upon excitation at 291 nm. At least three random fields were visualized for each biofilm, and representative images were displayed.

## Statistical analysis

All experiments in this study were conducted thrice independently. Each biological replicate was needed to conduct two technical replicates. Data are expressed as mean ± standard deviation (SD). Results were analyzed using SPSS 8.0 software and Origin 8.5

**Table 1 Minimum inhibitory concentration of nine kinds of drugs and antibiotics against seven *P. rettgeri* strains.**

| Stains (#) | MIC (µg/mL) | | | | | | | | |
|---|---|---|---|---|---|---|---|---|---|
| | SAG | AMP | FEP | ETP | IPM | TOB | AMK | TZP | F |
| 1 | 7.8 | >=32 | >=64 | >=8 | >=16 | >=16 | >=64 | >=128 | 32 |
| 2 | 7.8 | >=32 | >=64 | >=8 | >=16 | >=16 | >=64 | >=128 | 32 |
| 3 | 7.8 | >=32 | >=64 | >=8 | >=16 | >=16 | 16 | 64 | 512 |
| 4 | 7.8 | >=32 | >=64 | >=8 | >=16 | >=16 | 8 | >=128 | 512 |
| 5 | 7.8 | >=32 | >=64 | >=8 | >=16 | >=16 | 8 | >=128 | 512 |
| 6 | 3.9 | >=32 | >=64 | >=8 | >=16 | 4 | <=2 | >=128 | 256 |
| 7 | 3.9 | >=32 | 16 | >=8 | >=16 | 8 | 4 | >=128 | 256 |

Note:
AMP (Ampicillin); FEP (Cefepime); ETP (Ertapenem); IPM (Imipenem); TOB (Tobramycin); AMK (Amikacin); TZP (Piperacillin/tazobactam); F (Nitrofurantoin).

statistics. Analysis of variance (ANOVA) was performed to determine any significant differences ($P \leq 0.01$).

# RESULTS

## MIC of SAG against *P. rettgeri*

According to the MIC results presented in Table 1, *P. rettgeri* manifested antibiotic resistance, as evidenced by the MIC of ampicillin (32 µg/mL), cefepime (16–64 µg/mL), ertapenem (8 µg/mL), imipenem (16 µg/mL), tobramycin (8–16 µg/mL), amikacin (2–64 µg/mL), piperacillin/tazobactam (64–128 µg/mL) and nitrofurantoin (32–512 µg/mL). SAG exhibited excellent antibacterial activity against *P. rettgeri* 1-7, with the MIC values of 7.8, 7.8, 7.8, 7.8, 7.8, 3.9 and 3.9 µg/mL, respectively.

## Effects of SAG on the growth kinetics of *P. rettgeri* cells

The biomass of *P. rettgeri* was decreased when treated with SAG at MIC in media (Fig. 1); however, cells treated with SAG at 2 MIC exhibited a significant decline in the number of viable cells, which was lower than the detection limit after 24 h. However, the growth curves exhibited weaker increases and lower growth rates at concentrations of 1/8 and 1/4 MIC than those at other MIC. Higher concentrations of SAG resulted in a significant reduction in the relative number of viable cells, while the opposite has been found in lower SAG concentrations.

## SAG treatment decreased intracellular ATP concentrations, $pH_{in}$ and membrane potential of *P. rettgeri*

When *P. rettgeri* cells were treated with SAG at 1 and 2 MIC, the intracellular ATP concentration was significantly decreased ($P < 0.01$) in the SAG-treated group compared with the untreated control group (Fig. 2A); however, there were no significant differences between the cells treated with SAG at 1 and 2 MIC, but the decrease in intracellular ATP concentration was positively followed by the increase in SAG concentration.

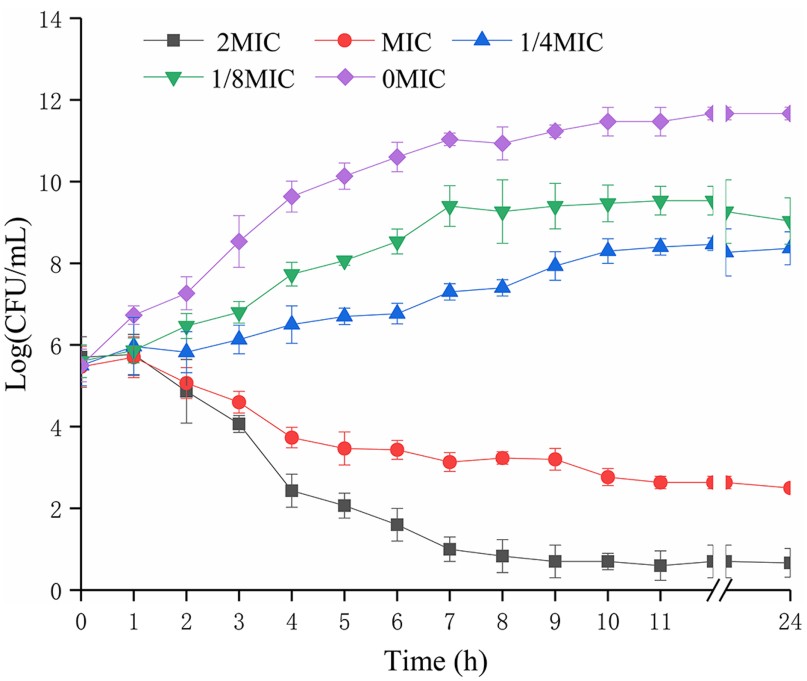

**Figure 1 Effects of SAG on the growth kinetics of *P. rettgeri* cells.** Effect of SAG on the growth of *P. rettgeri*. Bacterial cells were incubated and grown in TBS with 0, 1/8, 1/4, 1 and 2 MIC of SAG at 37 °C. Error bars are SD of three replicates.

Simultaneously, the extracellular ATP concentrations of *P. rettgeri* cells increased in a concentration-dependent manner, and its action was in a time-dependent manner (Fig. 2B).

The $pH_{in}$ of *P. rettgeri* after SAG treatment changed significantly, as presented in Fig. 2C. In this study, the original $pH_{in}$ of *P. rettgeri* was 5.97 ± 0.25, and treatment with SAG at 1 and 2 MIC decreased this $pH_{in}$ significantly to 4.53 ± 0.25 ($P < 0.01$) and 3.47 ± 0.25 ($P < 0.01$), respectively.

Moreover, the fluorescence intensities of *P. rettgeri* treated with SAG at 1 MIC for 2 h were less than those of the untreated group (Fig. 2D). Consistently, with an increasing concentration of SAG from 1 to 2 MIC, the membrane potential decreased significantly ($P < 0.01$).

## SAG treatment increased the cell membrane permeability of *P. rettgeri*

The cells with intact membranes stained with CFDA-SE exhibited bright green fluorescence, but membrane-damaged cells loaded with PI exhibited red fluorescence. In Fig. 3, *P. rettgeri* cells of the untreated control group exhibited bright green fluorescence, thereby revealing the physical integrity of the cell membrane. However, the results demonstrated significantly reduced green fluorescence and increased red fluorescence when treated with 1 MIC of SAG. As the SAG concentrations increased, green fluorescence declined, and red fluorescence gradually increased.
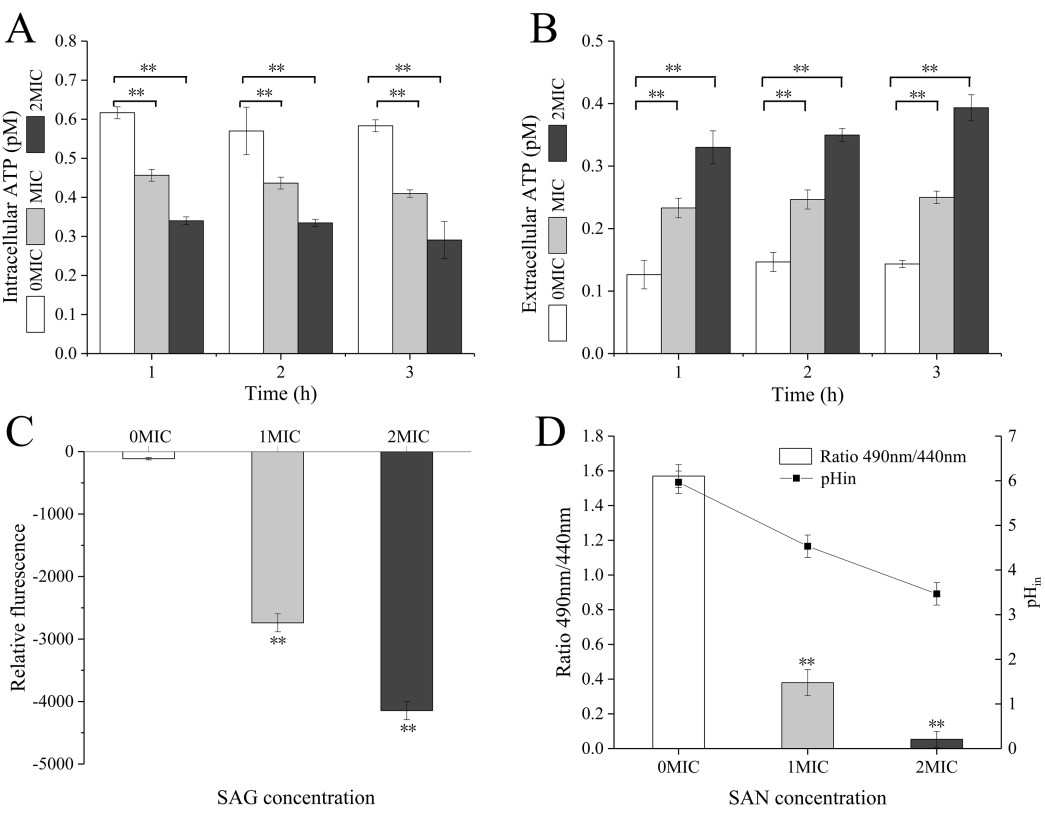

**Figure 2 SAG treatment led to a decrease in the intracellular ATP concentrations, pH and membrane potential of *P. rettgeri*.** Effects of SAG on *P. rettgeri*: (A) intracellular and (B) extracellular ATP levels, (C) membrane potential, and (D) pH$_{in}$. Data are expressed as mean ± SD. **$P < 0.01$ vs. 0 MIC.

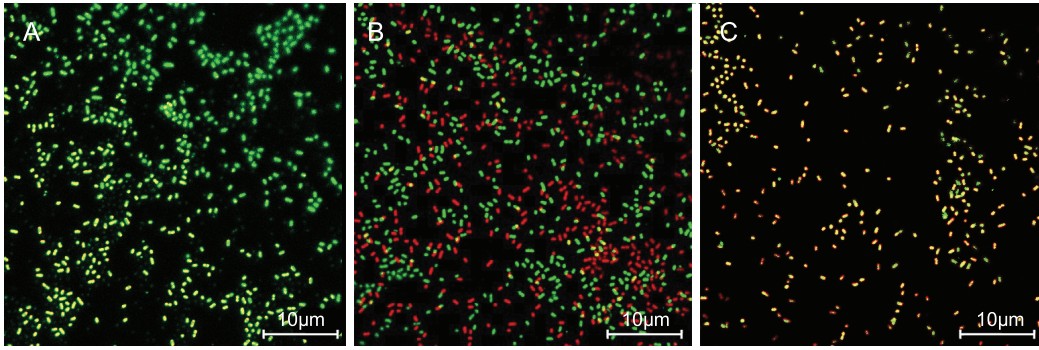

**Figure 3 SAG treatment increased the cell membrane permeability of *P. rettgeri*.** Effects of SAG on the cell membrane integrity of *P. rettgeri* cells through CLSM: (A) *P. rettgeri* cells exposed to 1% DMSO; (B) *P. rettgeri* cells exposed to SAG at 1 MIC, and (C) *P. rettgeri* cells exposed to SAG at 2 MIC.

## Treatment with SAG led to changes in the cell morphology of *P. rettgeri*

In this study, TEM was used to assess the level of cell wall damage and intracellular modification in SAG-treated *P. rettgeri*. The *P. rettgeri* cells without treatment exhibited visible outline and the peptidoglycan layer (Fig. 4A), while *P. rettgeri* cells treated with

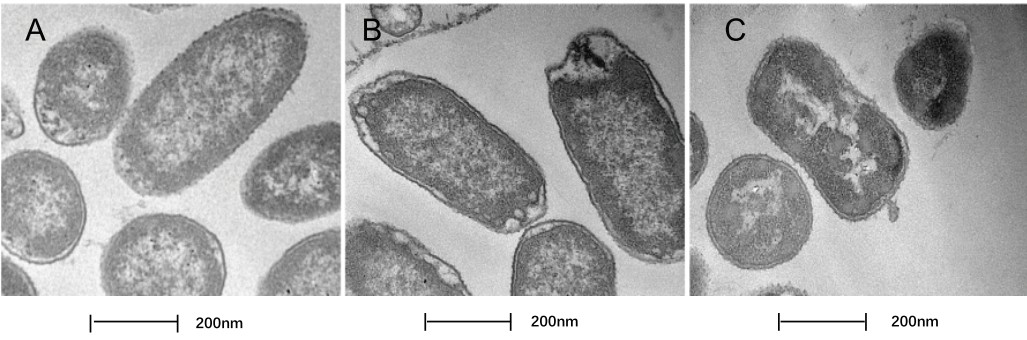

**Figure 4 Treatment with SAG led to changes in the cell morphology of *P. rettgeri*.** Effects of SAG on the cell structure of *P. rettgeri* through TEM. (A) *P. rettgeri* cells treated with 1% DMSO; (B) *P. rettgeri* cells treated with SAG at 1 MIC, and (C) *P. rettgeri* cells treated with SAG at 2 MIC.

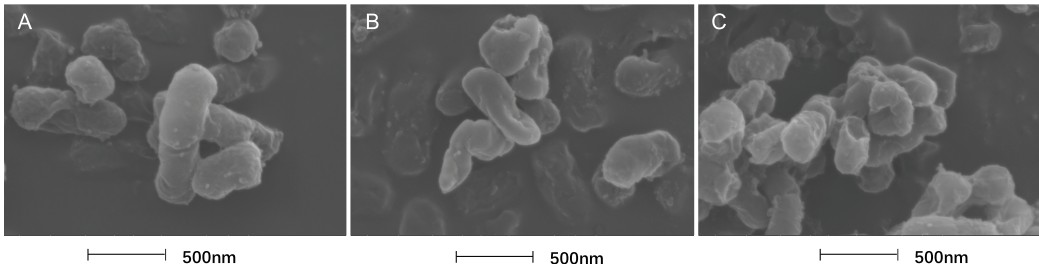

**Figure 5 Treatment with SAG led to changes in the cell morphology of *P. rettgeri*.** Effects of SAG on the cell morphology of *P. rettgeri* using FESEM: (A) *P. rettgeri* cells exposed to 1% DMSO; (B) *P. rettgeri* cells exposed to SAG at 1 MIC, and (C) *P. rettgeri* cells exposed to SAG at 2 MIC.

SAG at 1 MIC were malformed, damaged, and out of proportion (Fig. 4B). However, *P. rettgeri* cells treated with SAG at 2 MIC showed a wavy contour of the cytoplasmic membrane and dense, undifferentiable cellular content, indicating an obvious shrinkage of the cytoplasm (Fig. 4C). Consequently, SAG-treated *P. rettgeri* could damage the cell membrane and cell wall outline.

The cell morphology after treatment with SAG presented significant changes using FESEM. Compared with the normal smooth cell surface of the untreated group (Fig. 5A), cells treated with SAG at 1 MIC exhibited significant enlargement, uneven size, and rough surface (Fig. 5B). In addition, cells treated with SAG at 2 MIC exhibited significant surface collapse and expansion on the cell membrane as a result of the destruction of the bacterial cell wall (Fig. 5C), which indicated a positive correlation between the increase in the concentrations of SAG and the damage degree of cell membrane.

## Inhibitory effect of SAG on the biofilm formation of *P. rettgeri*

The crystal violet assay, FESEM and CLSM were used to analyze the inhibitory effects of SAG on the biofilm formation of *P. rettgeri*. As presented in Fig. 6, SAG exhibited a significant inhibitory effect on the biofilm formation of *P. rettgeri* at different

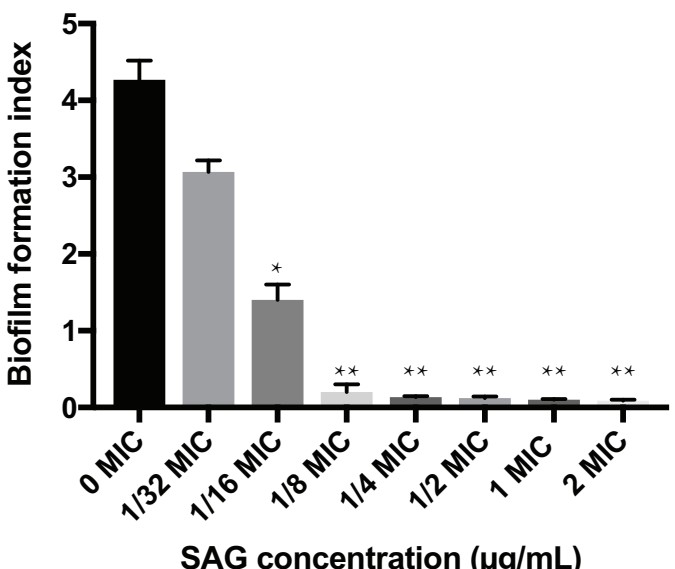

**Figure 6 Inhibitory effects of SAG on the biofilm formation of *P. rettgeri*.** The biofilm formation index was tested by crystal violet staining with different concentrations of SAG in 96-well plates. Data are expressed as mean ± SD. *$P < 0.05$ vs. 0 MIC, **$P < 0.01$ vs. 0 MIC. 

concentrations ($P < 0.01$). The biofilm formation index of the untreated control group was approximately 3.6. Biofilm formation was inhibited by 68% in the presence of 1/16 MIC SAG; furthermore, when the SAG concentration was higher than 1/4 MIC, the biofilm formation in *P. rettgeri* was suppressed by more than 95%. The results of FESEM and CLSM further showed that biofilm formation was significantly inhibited at 1/16, 1/8 and 1/4 MIC in the SAG-treated group compared with the untreated group (Fig. 7).

## Inactivation effect of SAG on the biofilm of *P. rettgeri* cells

The inactivation effect of SAG against 36-h-old biofilm-associated *P. rettgeri* cells is presented in Fig. 8. The untreated group was almost entirely green, as observed with CLSM (Figs. 8A and 8E), indicating that most of the cell membranes of *P. rettgeri* cells embedded in biofilms were integrated and viable. However, when *P. rettgeri* cells within biofilms were exposed to 16 MIC SAG, the CLSM observation displayed abundant red florescence, revealing that most of the cell membranes of *P. rettgeri* within biofilms were impaired, as presented in Figs. 8D and 8H. Furthermore, with an increasing concentration of SAG from 4 to 8 MIC, the green signal on the membrane disappeared gradually and the red signal increased gradually (Figs. 8B, 8C, 8F and 8G), indicating that intact and viable biofilms changed with the increasing concentration of SAG. These results suggested that SAG presented the killing effect on the *P. rettgeri* cells within biofilms.

## SAG could change the biofilm matrix composition of *P. rettgeri* cells

The biofilm matrix commonly comprises proteins, nucleic acid (environmental DNA (eDNA)) and carbohydrates, which provide structural rigidity and protection from the external environment to control gene regulation and nutrient adsorption (*Hobley et al., 2015*).

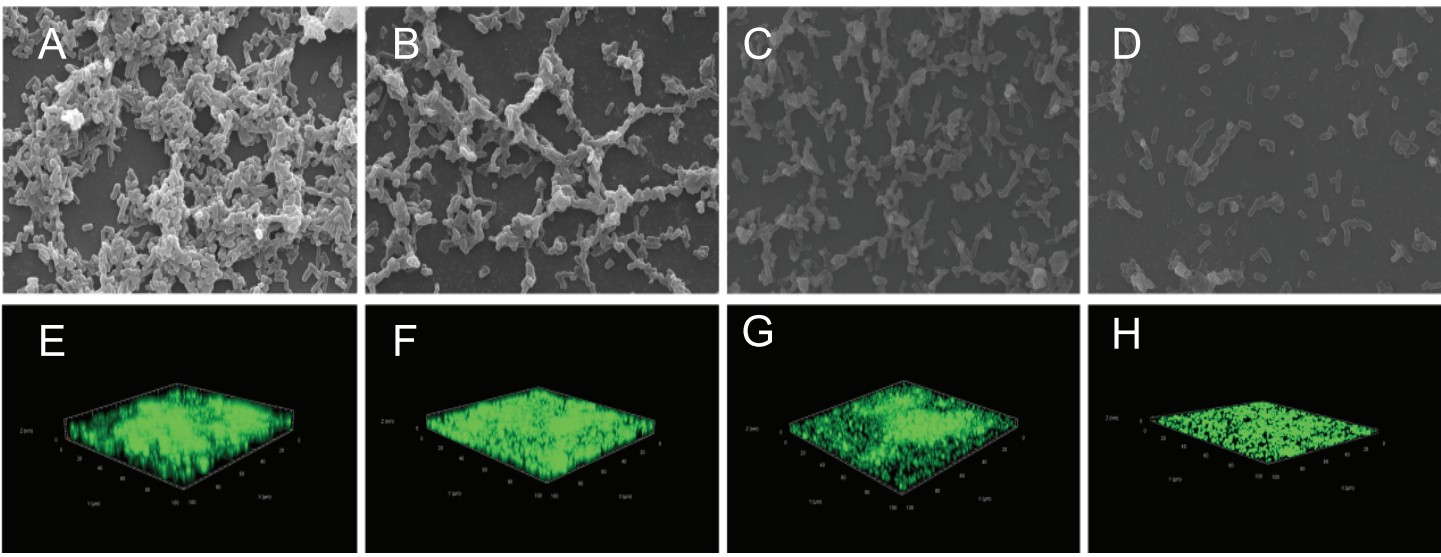

**Figure 7 Effect of SAG on the biofilm formation of *P. rettgeri*.** Images of FESEM (A–D; magnification, 10,000′×*g*) and CLSM (E–H). (A and E) *P. rettgeri* cells exposed to 1% DMSO; (B and F) *P. rettgeri* cells exposed to SAG at 1/16 MIC; (C and G) *P. rettgeri* cells exposed to SAG at 1/8 MIC, and (D and H) *P. rettgeri* cells exposed to SAG at 1/4 MIC.

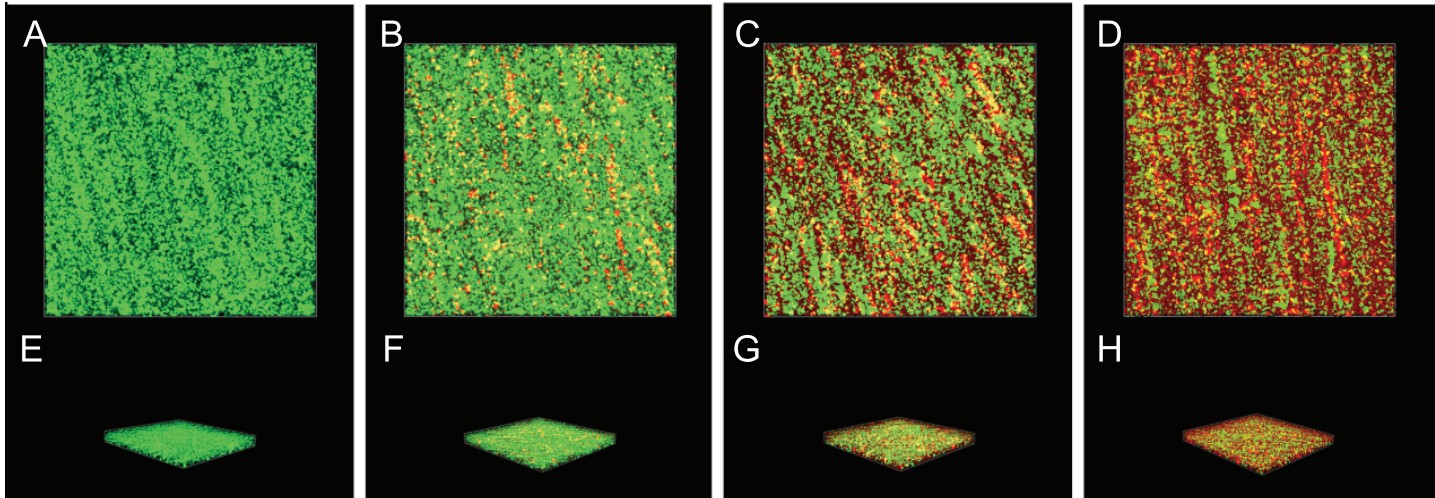

**Figure 8 Inactivation effect of SAG on the biofilm of *P.rettgeri* cells.** Inactivation effect of SAG on *P. rettgeri* cells within biofilms. 1D (A–D) and 3D (E–H) images of CLSM. (A and E) *P. rettgeri* cells within biofilms unexposed to SAG. (B and F) *P. rettgeri* cells within biofilms exposed to SAG at 1 MIC. (C and G) *P. rettgeri* cells within biofilms exposed to SAG at 2 MIC. (D and H) *P. rettgeri* cells within biofilms exposed to SAG at 4 MIC.

Thus, the changes in the biofilm matrix composition could affect biofilm formation. Therefore, the effects of SAG on the proteins, nucleic acid (eDNA) and carbohydrates of *P. rettgeri* were investigated. The changes in major biofilm matrix levels within 24-h-old biofilms of mono or dual species in the presence of SAG were detected by CLSM in Fig. 9. Different reagents were used to mark the eDNA and proteins red and the carbohydrates green. The untreated groups of nucleic acids and proteins exhibited almost red florescence, as observed with CLSM; with the increasing concentration of SAG, the red signal was

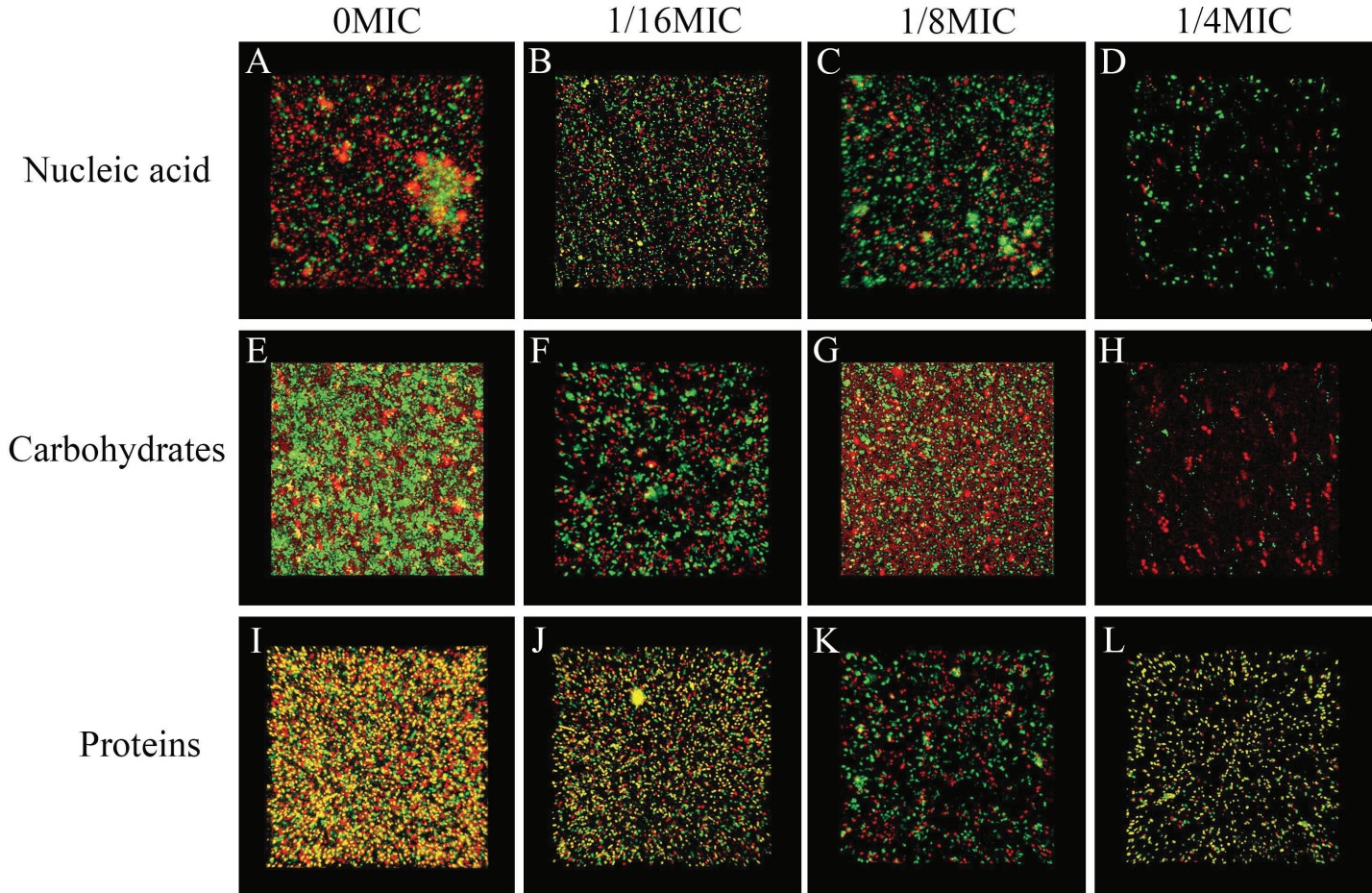

**Figure 9 SAG could change the biofilm matrix composition of *P. rettgeri* cells.** Effects of different concentrations of SAG on the levels of carbohydrates, extracellular proteins, and extracellular DNA inside *P. rettgeri* biofilms. (A–D) eDNA labeled red (PI). (E–H) Carbohydrates labeled green (WGA). (I–L) Proteins labeled red (SYPRO Ruby). Scale bar, 10 μm.

gradually reduced (Figs. 9A and 9C). The untreated group of carbohydrates exhibited green florescence, as observed with CLSM; with an increasing concentration of SAG, the green signal gradually disappeared (Fig. 9B), indicating that nucleic acid, protein, and carbohydrate contents decrease with an increasing concentration of SAG.

## Effects of SAG on biofilm diffusion

Gatifloxacin was used to verify the effects of SAG on biofilm diffusion, which was monitored using fluorescence CLSM. *Providencia rettgeri* biofilms were grown with different SAG concentrations for 48 h and then added with gatifloxacin for 4 h. Formed biofilms were stained with SYTO9 to allow the visualization of nucleic acid (green fluorescence).

Gatifloxacin diffused significantly at 0 MIC (Fig. 10A). However, in mixed biofilms with 1/4 MIC of SAG, the gatifloxacin signal was minimally detected (Fig. 10D). With an increasing concentration of SAG, the signal diffusion became weaker, thus proving that SAG inhibits biofilm formation.

0MIC                           1/16MIC

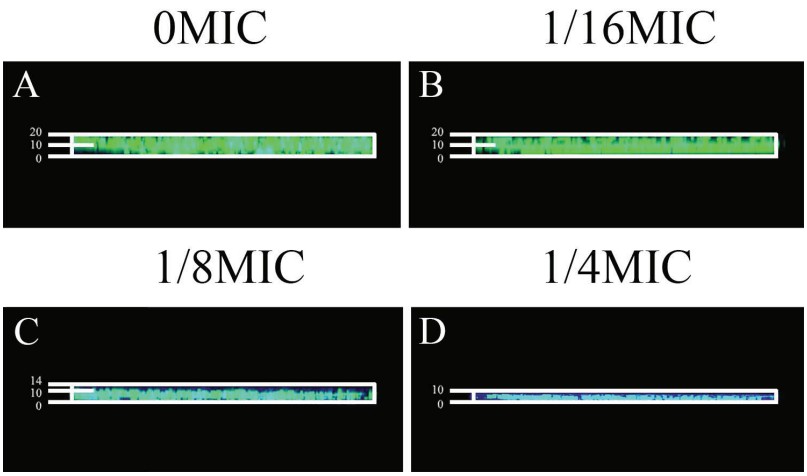

1/8MIC                          1/4MIC

**Figure 10 Effect of SAG on biofilms of diffusion.** (A) *P. rettgeri* cells treated with 0 MIC and gatifloxacin; (B) *P. rettgeri* cells treated with 1/16 MIC and gatifloxacin; (C) *P. rettgeri* cells treated with 1/8 MIC and gatifloxacin; (D) *P. rettgeri* cells treated with 1/4 MIC and gatifloxacin.

## DISCUSSION

To the best of our knowledge, the present study is the first to evaluate the antibacterial activity and mechanism of SAG against *P. rettgeri*. The findings of TEM, CLSM, and FESEM revealed significant effects of SAG against *P. rettgeri* biofilms, including its inhibitory effect on biofilm substance expression and formation as well as on biofilm inactivation. We found that SAG could decrease intracellular ATP concentration and changes in $pH_{in}$ and reduce the membrane potential.

Here SAG demonstrated strong activity against *P. rettgeri* biofilms. SAG not only inhibited biofilm formation but also destroyed the intact and viable biofilm. At 1/16 MIC, SAG inhibited biofilm formation by approximately 68%, whereas at 1/4 MIC, more than 95% of the biofilm was inhibited, thus an outstanding antibacterial effect of SAG was observed on *P. rettgeri*.

ATP depletion is a common biological change associated with cell damage, suggesting that ATP is used as a potential indicator of the effects of antimicrobial agents on the intact and viable cell membrane. These results presented that SAG-treated *P. rettgeri* exhibited a remarkable decrease in intracellular ATP concentrations, and changes in intracellular and extracellular ATP balance were observed in *P. rettgeri* cells, suggesting cell membrane damage. These results were generally consistent with the results of other analyses (*Qian et al., 2020*).

Intracellular pH has a critical influence on cell physiology, including DNA replication, RNA transcription, protein synthesis and enzyme activity. Moreover, $pH_{in}$ is positively related to intact and viable cell membranes (*Gonelimali et al., 2018*). Consequently, the cell membrane damage can be reflected by the changes in the $pH_{in}$. In the present study, changes in the $pH_{in}$ of *P. rettgeri* were significantly related to SAG concentrations, wherein the addition of 2 MIC SAG led to a decrease in the $pH_{in}$ from 5.97 ± 0.25 to 3.47 ± 0.25.

Hence, as described in the previous literature for plant-derived products, the effect of SAG on *P. rettgeri* could be cleared as the breaking to intact and viable cell membrane. In addition, when SAG entered into cells, it could cause hyperpolarization across the cell membrane, thus making the internal membrane potential more negative. Compared with that in the untreated control group, when the SAG concentration increased from 1 to 2 MIC, the membrane potential significantly decreased in the SAG-treated group. Similarly, *Zhong et al. (2017)* reported that SAG could decrease the membrane potential of *Candida albicans* cells. Taken together, these results suggest that SAG causes plasma membrane hyperpolarization fin determined bacteria, which possibly resulting in cell metabolic activity disruption and cell death.

Our studies have also indicated that *P. rettgeri* cells exhibit significant membrane dissolution, as observed with CLSM, whereas leakage of cytoplasmic components has been reported after exposure to SAG, as observed with TEM. Herein, CLSM revealed that the permeability of cell membrane increased after treatment with SAG. The 2 MIC-exposed *P. rettgeri* cells were apparently entirely red, indicating that the majority of cell membranes were destroyed and that some substances could move allodially in and out of the cell. Therefore, FESEM images of *P. rettgeri* cells exposed to SAG at 2 MIC revealed that the cells had severe morphological alterations, whereas MIC-exposed cells were visibly extended and had a rough surface. The physical and morphological changes in *P. rettgeri* cells may be due to the effect of SAG on membrane permeability and integrity. However, dense undifferentiated intracellular materials in SAG-exposed cells were observed with TEM. Moreover, CLSM revealed that almost all bacterial cells were damaged after exposure to SAG at 2 MIC. As reported by *Matijašević et al. (2016)*, FESEM and TEM were used to observe the morphological changes in *S. aureus* cells treated with *Coriolus versicolor* methanol extract.

Previous studies have verified that biofilms could enhance bacterial resistance to adverse environmental pressures, including resistance to antibiotics and antimicrobial agents (*Jakobsen, Tolker-Nielsen & Givskov, 2017*). We found that exopolysaccharides, proteins and extracellular DNA formed bacterial cells in biofilms, which were embedded in extracellular polymeric substances. In this study, the effects of SAG on biofilm formation and biofilm-associated cell inactivation was further illustrated, and the results of crystal violet staining revealed that SAG at different concentrations presented inhibitory effects on the biofilm formation of *P. rettgeri*. Moreover, FESEM and CLSM results illustrated that SAG-exposed cells exhibited substantially decreased adhesion and survival, indicating that biofilm formation was significantly inhibited by SAG. These results were in agreement with the effectiveness of antistaphylococcal lysin CF-301 on the inactivation of 95 *S. aureus* strains within biofilms, as described by *Schuch et al. (2017)*. In addition, CLSM further revealed that SAG exhibited robust impaired effects on biofilm-associated *P. rettgeri* cells. CLSM images of *P. rettgeri* cells within biofilms exposed to 4 MIC of SAG displayed dark red florescence, proving that most of the membranes of cells within the biofilm were damaged. Our study were divided into two main parts, one part demonstrated SAG had antimicrobial activity against *P. rettgeri* through causing cell membrane dysfunction and changes in cellular morphology, similarlgy a study by *Qian et al. (2020)*,

and the other part was the effect of SAG on *P. rettgeri* biofilm, which showed that different concentrations of SAG could inhibit the formation of *P. rettgeri* biofilm by changing the biofilm matrix while also killing through mature biofilms in chronic infections. The results were in line with previous reports that showed that SAG inhibits biofilm formation of *Candida albicans* and *Staphylococcus aureus* and synchronously reducing extracellular proteins, polysaccharides, and eDNA levels in a dose-dependent manner (*Qian et al., 2020*). These findings revealed that SAG exhibits potent antibacterial and antibiofilm activity against *P. rettgeri*, and thus has potential to be exploited as a natural preservative to control the *P. rettgeri* associated infections.

## CONCLUSIONS

In the present study, we confirmed the efficacy of SAG in inactivating both the biofilm formation and integrity of *P. rettgeri* cells. SAG could induce cell lysis, leading to cell membrane damage and intracellular component leakage in *P. rettgeri* cells. In addition, 2 MIC SAG could inactivate the biofilm formation in *P. rettgeri* cells, indicating its potential as a natural antibacterial agent to control the negative impact of *P. rettgeri* in the food industry.

### Funding
This work was supported by the National Natural Science Foundation of China (Nos. 81902067 and 81300028). The funders had no role in study design, data collection and analysis, decision to publish, or preparation of the manuscript.

### Grant Disclosures
The following grant information was disclosed by the authors:
National Natural Science Foundation of China: 81902067 and 81300028.

### Competing Interests
The authors declare that they have no competing interests.

### Author Contributions
- Qian Zhang conceived and designed the experiments, prepared figures and/or tables, authored or reviewed drafts of the paper, and approved the final draft.
- Yansi Lyu performed the experiments, prepared figures and/or tables, and approved the final draft.
- Jingkai Huang performed the experiments, prepared figures and/or tables, and approved the final draft.
- Xiaodong Zhang analyzed the data, prepared figures and/or tables, and approved the final draft.
- Na Yu analyzed the data, prepared figures and/or tables, and approved the final draft.
- Ziping Wen analyzed the data, prepared figures and/or tables, and approved the final draft.

- Si Chen conceived and designed the experiments, prepared figures and/or tables, authored or reviewed drafts of the paper, and approved the final draft.

## Data Availability

The raw data are available in a Supplemental File.

## Supplemental Information

Supplemental information for this article can be found online at http://dx.doi.org/10.7717/peerj.9543#supplemental-information.

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
