# Peer review of "Antibacterial activity and mechanism of sanguinarine against Providencia rettgeri in vitro"

_PeerJ, doi:10.7717/peerj.9543_

## Round 0.1 · original submission · Major Revisions

Please address the issues highlighted by the reviewers. I have also found two problems in your figures: The legend to Figure 7 seems to belong to a different figure (since it only labels 6 panels instead of the actual 8 panels present in Fig.7). Fig. 10 is hardly legible: please consider stacking the 4 panels vertically instead of horizontally, or at least dispose them in a 2*2 grid.

Reviewer 1 ·

Basic reporting

The manuscript describes antibacterial activity and mechanism of sanguinarine against Providencia rettgeri. The manuscript presents potential valuable information on novel antibiotics, but it suffers from some lack of consideration of some aspects. Major revision is needed.
Major comments:
1. The results revealed that the MIC of SAG against P. rettgeri was 7.8 μg/mL. Accurately, i do not think this is a strong antibacterial effect. Do authors have done experiment of SAG acting on other bacteria? Or any references?
2. This manuscript must be edited for proper grammar and overall style. For examples, the sentence of line 58-59 is too elusive to read. The subsequent sentences (line60-62) also presents grammar problems.

Experimental design

The authors only mentioned Sanguinaria was purchased. Was SAG extracted from Sanguinaria? The key process was missed in 'Materials and Methods'.

Validity of the findings

Line 73, authors mentioned 'this study aimed to investigate the antimicrobial and antibiofilm activities of Sanguinaria against P.rettgeri'. Is SAG equal to Sanguinaria?

Additional comments

1. Line 64-65, SAG is extracted from the rhizome of Sanguinaria canadensis. In fact, SAG can be extracted from Chelidonium majus, Macleaya cordata, et al. Please find more references and revise this paragraph.
2. Line 222-225, which specific statistical method was used in MS?
3. Line 30-31, Whether decrease in intracellular ATP could definitely cause damage of the cell membrane?
4. The authors did several experiments of SAG on P. rettgeri. However, each experiment seemed independent and unconnected. Authors need to integrate all experimental results and explain how SAG inhibite P. rettgeri in 'Discussion'.
5. Figure 2 need to recreate, especially 2A and B.

·

Basic reporting

The overall expression of this article is clear, but the words or sentences in many places are not professional and fluent. Please seek a native English translator or professional translation service to help modify the language of this article.
Given that this article is about using sanguinarine as a potential food preservative, by studying its antibacterial activity and antibiofilm activity, the safety of SAG is not mentioned in the introduction, which limits the significance of the research.
Some references are incorrectly marked in the article, please carefully find and correct them.

Table 1 is missing the MIC experimental data of SAG, please add.

Experimental design

The experimental design of the article is very comprehensive. It not only studies the antibacterial activity and mechanism of SAG, but also the antibiofilm activity of SAG, which provides extensive data for its application as a food preservative. However, the description of the MIC experiment is not accurate enough. Choi et al used the disc diffusion method to measure the MIC and you used the microbroth dilution method. This is the wrong reference, and why is there no reference for the standard MIC measurement method (Clinical and Laboratory Standards Institute. 2012. Methods for dilution antimicrobial susceptibility tests for bacteria that grow aerobically; approved standard (9th Edition). Wayne: Clinical and Laboratory Standards Institute)?

Validity of the findings

None

Additional comments

1. line 30, "The gradient diffusion method" is an inaccurate expression,It should be microbroth dilution method according to MIC test.
2. lines 53,182, 370, 379, Incorrect reference notes.
3. line 78, Please provide the city and country of the reagent manufacturer used in this article.
4. lines 79-81, After ten-fold dilution, the SAG concentration was 1 mg / ml and the dmso was 100% v / v. It is not difficult to infer that the highest concentration in the MIC experiment was 500 ug / ml, and the dmso concentration was 50%. In fact, only when the dmso concentration is lower than 2%, the bactericidal effect will not be affected. Have you considered the effect of DMSO on the antibacterial effect?
5. line 94, The bacterial concentration is 10^2 CFU/ml and the reference is 5x10^6 cfu / ml. Why is it diluted to this concentration? The dilution is likely to affect the result of the MIC measurement.
6.line 96, replaced "visible" by "viable".
7. line 102, "1:8, 1:4, 1:2", 1/8,1/4,1/2 are more appropriately.
8. line 109, 163, Reference citation malformed.
9. line 167, repalced "alcohol" by "ethanol".
10. line 233, delete "Table1".
11. lines 233-234, "The results of the antibacterial activities ... against P. rettgeri." is useless description.
12. lines 234-235, "SAG was effective with an MIC of 31.25, 15.6, and 7.8 μg/mL against P. rettgeri." could be described as "SAG exhibited excellent antibacterial activity against P. rettgeri 1-8, with the MIC values of 31.2,..., respectively." for better.
13. line 242, What is "this" mean?
14. line 274, delete "exhibited".
15.lines 312-313, " eDNA and proteins red" As both eDNA and proteins were stained to red, how to separate them?
16. line 337, antifungal??
17. lines 342-345, "A similar study found... peppermint essential oil." What is the purpose of these citation? These reagents are not related to SAG. These are just quotations of others' results without further discussion!
18. lines 350-353, " A similar study by... to plant extracts." These are just quotations of others' results without further discussion!
19. line 361, Previous studies? not this paper?
20. line 377, observed ??
21. line 215, What is LUT and P.R cells?Do not be metioned above.
22. Figure 7, (A–C; magnification, 10,000×) and CLSM (D–F; 3D). Wrong label, A-D for FESEM; E-H for CLSM.

---

## Round 0.2 · Minor Revisions

Please address the final questions from Reviewer 2.

Reviewer 1 ·

Basic reporting

no comment

Experimental design

no comment

Validity of the findings

no comment

·

Basic reporting

no comment

Experimental design

no comment

Validity of the findings

no comment

Additional comments

Although this manuscript has been revised thorough, the following issues should be further clarified before publication.

1. Materials & Methods Minimum inhibitory concentration (MIC) test, the authors used the agar dilution method to determine the MIC by mixing the bacterial samples and SAG into 96-well plate rather than 9-cm Petri dishes according to the reference. As we know, 96-well plate is normally used in the broth microdilution and 9-cm Petri dishes is used in the agar dilution. Why do you choose the 96-well plate in this study?
2. In the previous manuscript, the bacterial concentration used in MIC test is 10^2 CFU/ml. You've changed it to 106 CFU/mL in the revised manuscript (line 105), but without any description of repetitive experiment for the influence of bacterial concentration on the MIC values. So the results of the MIC test are strongly doubtful.
3. Line 432, “Inhibits” should be “inhibits”, and “Candida albicans and Staphylococcus aureus” should be italicized.
4. Caption of Figure 1, replace “1:8, 1:4” with 1/8, 1/4.

---

## Round 0.3 · accepted · Accept

Thank you for addressing the remaining issues. I look forward to seeing your manuscript in print!